# TSR-YOLO: A Chinese Traffic Sign Recognition Algorithm for Intelligent Vehicles in Complex Scenes

**DOI:** 10.3390/s23020749

**Published:** 2023-01-09

**Authors:** Weizhen Song, Shahrel Azmin Suandi

**Affiliations:** Intelligent Biometric Group, School of Electrical and Electronics Engineering, University Sains Malaysia, Engineering Campus, Nibong Tebal 14300, Malaysia

**Keywords:** traffic sign, intelligent vehicle, YOLOv4-tiny, k-means++, CCTSDB2021 dataset

## Abstract

Recognizing traffic signs is an essential component of intelligent driving systems’ environment perception technology. In real-world applications, traffic sign recognition is easily influenced by variables such as light intensity, extreme weather, and distance, which increase the safety risks associated with intelligent vehicles. A Chinese traffic sign detection algorithm based on YOLOv4-tiny is proposed to overcome these challenges. An improved lightweight BECA attention mechanism module was added to the backbone feature extraction network, and an improved dense SPP network was added to the enhanced feature extraction network. A yolo detection layer was added to the detection layer, and k-means++ clustering was used to obtain prior boxes that were better suited for traffic sign detection. The improved algorithm, TSR-YOLO, was tested and assessed with the CCTSDB2021 dataset and showed a detection accuracy of 96.62%, a recall rate of 79.73%, an F-1 Score of 87.37%, and a *mAP* value of 92.77%, which outperformed the original YOLOv4-tiny network, and its FPS value remained around 81 f/s. Therefore, the proposed method can improve the accuracy of recognizing traffic signs in complex scenarios and can meet the real-time requirements of intelligent vehicles for traffic sign recognition tasks.

## 1. Introduction

Traffic sign recognition is a crucial component of intelligent vehicle driving systems and one of the most important research fields in computer vision [1]. Traffic sign recognition tasks are usually performed in natural scenes; however, extreme weather conditions (e.g., rain, snow, or fog) can obscure traffic signage information, and overexposure and dim light usually reduce the visibility of traffic signs. Furthermore, traffic signs are exposed all year, causing the surfaces of some to fade, become unclear, or become damaged. Complex and changing environments often affect the speed and accuracy of traffic sign recognition in intelligent transportation [2]. Therefore, it is now especially essential to study the problem of fast and accurate traffic sign detection in complex environments.

Early recognition methods in traffic sign recognition used a sliding window strategy to traverse the entire image and generate many candidate regions. The candidate regions were then extracted with various types of hand-designed features, such as HOG (histogram of oriented gradient) [3], SIFT (scale-invariant feature transform) [4], and LBP (local binary pattern) [5]. These features were then fed into an efficient classifier, such as SVM (support vector machine) [6], Adaboost [7], or Random Forest [8], for detection and identification. However, traditional target detection methods require researchers to extract features manually and are not robust to changes in diversity. In addition, sliding-window-based region selection strategies are not targeted and have high time complexity. Hu et al. [9] proposed a new approach for traffic sign detection based on maximally stable extremal regions (MSERs) and SVM, which had a high level of accuracy but only seven frames per second (FPS) of detection speed. Dai et al. [10] proposed using color to improve the recognition rate of traffic signs in varying brightness environments, achieving 78% accuracy and 11 FPS. Nevertheless, in real scenarios, real-time and accuracy are essential for traffic sign recognition. Therefore, conventional methods of target detection fall well short of the needs of intelligent traffic systems.

The AlexNet [11] algorithm achieved great success for convolutional neural networks in 2012, making deep learning rapidly gain the attention of researchers in the field of artificial intelligence, including target detection. Girshick et al. proposed R-CNN (regions with CNN features) [12], the first deep-learning-based two-stage target detection algorithm, which provided a significant performance improvement compared to traditional algorithms. The algorithms that followed, such as SSD (single shot multi-box) [13], Fast R-CNN [14], Faster R-CNN [15], and the YOLO (you only look once) series [16,17,18,19], achieved higher accuracy in target localization and classification tasks. Zhang et al. [20] proposed the MSA_YOLOv3 algorithm for traffic sign recognition, with a *mAP* value of 0.86 and a detection speed of 9 FPS. Zhang et al. [21] proposed CMA R-CNN for traffic sign recognition, with a *mAP* value of 0.98 and a detection speed of only 3 FPS. Cui et al. [22] proposed CAB-s Net for traffic sign detection, with a *mAP* value of 0.89 and a detection speed of 27 FPS. However, these algorithms are frequently designed to extract more detailed features by constructing deeper network structures, resulting in models that are relatively large, are slow to detect, and require high amounts of hardware computing power and storage capacity, making them difficult to use in mobile and embedded devices.

In order to accelerate the detection time of deep convolutional neural-network-based traffic sign detection methods, a lightweight convolutional neural-network-based target detection architecture is now used to recognize traffic signs. Regarding detection speed, YOLOv4-tiny [23] is a superior target detection model that outperforms the vast majority of current, complicated deep convolutional neural network models. However, the YOLOv4-tiny algorithm’s detection accuracy is relatively low. This paper proposes a Chinese traffic sign detection algorithm based on enhanced YOLOv4-tiny that can more effectively promote the transmission and sharing of different levels of information to improve the algorithm’s detection accuracy and ensure its detection speed by optimizing the network. Compared to the YOLOv4-tiny algorithm, the following are the primary contributions of this study.

To address the issue that a complex background interferes with target recognition in the feature information extracted by the CSPDarknet53-tiny network, this paper embeds a BECA attention mechanism module in a CSP structure to improve the model’s ability to extract and utilize key feature information while reducing the importance of useless features and to invest computational resources in different channels proportionally to the importance of the channels.Since the YOLOv4-tiny enhanced feature extraction network is too simple, and the fusion of feature layers only reflects the stacking of a single feature layer after upsampling, resulting in a low utilization of feature information extracted from the backbone network and insufficient feature fusion, dense spatial pyramid pooling (Dense SPP) is introduced for multiscale pooling and the fusion of input feature layers to enrich the feature expression capability.Based on the original network, the detection scale range is increased to improve the degree of matching for targets of various sizes. The bottom–up fusion of deep semantic information with shallow semantic information is used to improve the feature information of small targets, predict small and far away traffic sign targets more accurately, and improve the accuracy of the network’s localization and detection.In order to accelerate the network’s ability to detect traffic signs, k-means++ clustering is used to learn prior boxes that are more suitable for traffic sign detection.The TSR-YOLO method proposed in this study has a higher *mAP* value of 8.23%, a higher precision value of 5.02%, a higher recall value of 1.6%, and a higher F-1 score of 3.04% compared to YOLOv4-tiny.

The rest of this paper is organized as follows. In Section 2, we briefly review the development of target detection, traffic sign detection, and related work. Section 3 describes our research methodology. Section 4 presents our experimental results and analysis. Section 5 summarizes our work and provides some suggestions for future work.

## 2. Related Work

Traffic sign detection is one of the most challenging and essential problems in autonomous vehicle-driving systems. Most early algorithms for identifying traffic signs used machine learning and template matching [24]. Deep-learning-based algorithms are widely used for high-precision traffic sign detection due to the rapid development of high-performance computers and the enormous explosion of data volume in recent years.

Tong et al. [25] proposed a color-based support vector machine (SVM) algorithm for traffic sign recognition that first converted the RGB color space to HSV color space to determine the region of interest (ROI) and then extracted the histogram of oriented gradients (HOG) features and used an SVM to determine whether it was a traffic sign. Yu et al. [26] identified traffic signs using a color threshold segmentation method and morphological processing to eliminate the interference of the background region and increase the contours of the sign region and then used the HOG method to gather the gradient of each pixel point within a cell. Madani et al. [27] employed adaptive thresholding algorithms and support vector machine models to recognize and classify traffic signs based on boundary color and shape.

Typically, the performance of such detection methods is dependent on the useful-ness of the manual feature extraction, which requires shape features, color features, or hybrid features to obtain rich detail information of traffic signs. Detection results are also susceptible to objective natural factors, such as variations in light, extreme weather, and obstructions.

Since the emergence of deep-learning techniques, numerous target detection algorithms based on deep learning have been applied to traffic sign detection [28]. In contrast to the above methods, deep-learning models can automatically extract features, avoiding the limitations of manual feature extraction, and their generalizability and robustness are relatively high. There are currently two types of CNN-based target identification algorithms: single-stage detectors based on regression and two-stage detectors based on candidate areas. Zuo et al. [29] used a two-stage target detection algorithm, Faster R-CNN, to detect traffic signs by conditionally scanning an image to generate a large number of candidate boxes, sending each candidate box to the network to extract a feature, sending that feature to a classifier for classification, and finally generating the correct class name. Li et al. [30] designed a detection model using the Faster R-CNN and MobileNet structures. It refined the localization of small traffic signs using color and shape information. A CNN with an asymptotic kernel was then used to classify traffic signs. The research results demonstrated that the proposed detector was able to detect different kinds of traffic signs. Unlike the two-stage target detection method, the single-stage target detection algorithm first uses a clustering algorithm to create a certain number of prior boxes. It then uses these prior boxes to find a region of interest, feeds the region of interest into a feature extraction network, and uses a regression method to determine the confidence probability of the object. This accelerates operation and allows for real-time detection. Shan et al. [31] used an SSD single-stage target identification method to detect traffic signs; the algorithm worked well with single-target, multi-target, and low-light images. Chen et al. [32] proposed employing the YOLOv3 method to overcome the problem of poor rate of traffic sign recognition due to complicated background interference and, ultimately, achieved accurate traffic sign recognition by fusing advanced network modules.

In conclusion, detection methods based on deep learning can enable intelligent vehicles to better detect traffic signs in complex road scenarios. With the rapid development of intelligent vehicles, real-time and accuracy requirements for traffic sign detection and recognition have improved. This paper employs a single-stage deep learning detection method and proposes TSR-YOLO, a lightweight traffic sign detection model with high accuracy, low latency, and robustness to improve detection performance.

## 3. The Proposed Method

This study creates an effective traffic sign identification algorithm and integrates the proposed TSR-YOLO model into a vehicle traffic sign perception system. This section begins with an overview of the smart car traffic sign recognition system, followed by a brief description of the YOLOv4-tiny network and a discussion of the YOLOv4-tiny network improvement method.

### 3.1. The Traffic Sign Recognition System

This study demonstrates an intelligent vehicle traffic sign visual perception system with three main parts: a vision system, a traffic sign detection system, and an intelligent car display system. To be more specific, a vision system based on a monocular camera captured information in a vehicle’s driving road environment in the form of video or image and then passed the information to a traffic sign detector, which detected the existence of traffic signs in the driving environment by the video sequence given by the vision system. If the traffic sign information was captured in the road environment, it was displayed on the HUD (head up display) flat-strip display. The responsibility of the traffic sign detection system was to detect the existence of traffic signs in the driving environment. It was a key component of the proposed system for identifying traffic signs. Therefore, efforts needed to be made to develop a system capable of detecting traffic signs rapidly and precisely in a complicated road environment. Figure 1 illustrates the proposed traffic sign recognition system’s workflow.

A traffic sign recognition system can effectively remind drivers to pay attention to traffic sign information, such as prohibitions and warnings, to prevent violations caused by negligence. In our study, a monocular camera captured video sequences in real time. The camera was the “eye” for traffic sign detection and was connected to a computer system running an improved YOLOv4-tiny pretraining model. If the pretrained detector detected information containing traffic signs in the road environment, it passed the information to an intelligent vehicle display system for display on the HUD.

### 3.2. The YOLOv4-Tiny Network

YOLOv4-tiny is a scaled-down version of YOLOv4. The main idea is to treat the target detection task as a regression problem, with the detected target location and classification results obtained directly through network model regression. Figure 2 depicts the network structure of YOLOv4-tiny. The YOLOv4-tiny network is divided into three components: the backbone (CSP-Darknet53-tiny), the neck (feature pyramid network, FPN), and the Yolo-head. (1) The backbone part is composed of a convolutional block (CBL), a maximum pooling layer (maxpool), and a cross-stage partial (CSP) module, which is mainly used for prefeature extraction. (2) In the neck part, YOLOv4-tiny retains the feature pyramid network (FPN) structure of YOLOv4. The FPN structure can fuse the features between different network layers so that it can obtain both the rich semantic information of the deeper networks and the geometric detail information of the lower networks to enhance the feature extraction ability. (3) Two prediction branches are retained in the Yolo-head section, and the final prediction is performed using the feature fusion results obtained from the FPN module to form two prediction scales of 13 × 13 and 26 × 26. Because of its simple structure, small computation, and fast detection time, YOLOv4-tiny is suitable for intelligent vehicle environment-aware systems. Still, it is not very accurate in detecting small targets, such as traffic signs, which makes it difficult to adapt to the task of traffic sign recognition in complex scenes. Therefore, some improvements to YOLOv4-tiny are needed to make the algorithm capable of detecting traffic signs in complex scenarios.

### 3.3. The Proposed TSR-YOLO Algorithm

For the specific traffic sign detection task, we improved the YOLOv4-tiny algorithm’s ability to extract features by adding an improved BECA attention mechanism module to a CSPDarknet53-tiny structure, combining an improved spatial-pyramid-pooling module with the FPN structure and adding a Yolo detection layer to the Yolo head. The CCTSDB2021 traffic sign dataset was grouped using the k-means++ algorithm to find the anchor boxes that the model used.

#### 3.3.1. The Improvement of CSPDarknet53-Tiny

A color picture has three channels of RGB. After convolution by different convolution kernels, each channel produces new channels. The new channels’ features reflect the image components on distinct convolutional kernels, which do not contribute equally to the task’s crucial information. The performance of a network can be improved by blocking out irrelevant information and giving important information a higher weight value. In 2019, Hu et al. [33] proposed the SENet channel attention mechanism, which significantly enhanced the performance of convolutional neural network models. ECANet [34] is an improved lightweight channel attention mechanism compared to the SENet module. Global averaging pooling is performed before processing the features. Global averaging pooling sums and averages all weights of the same channel, which results in some high and low weights being averaged and a loss of information about the high weights. As a result, in this paper, we used Better-ECA [35], an improved ECA attention mechanism that incorporated maximum global pooling (BECA). Figure 3 depicts the improved BECA channel attention mechanism’s structure.

Feature compression

In this step, global average pooling was utilized to compress the input H × W × C features into 1 × 1 × C features W, while maximum global pooling was used to extract the maximum value of the channels to produce 1 × 1 × C features U. The features acquired in the two parts were then subjected to a fusion operation, and their channel information on the corresponding channels was summed as shown in Equation (1):(1)$$Z_{c}=W_{i}+U_{i}$$
where $W_{i}$ is the feature information of the global average pooling channel, and $U_{i}$ is the feature information of the global maximum pooling channel.

2.Characteristic incentive

A one-dimensional convolution with a convolution kernel of size k captured only the k-neighboring channels of the input features instead of all the channels. This could significantly reduce the parameters and computational costs. The convolved features were then activated by the sigmoid activation function to output the feature information of each channel, where the operation could be represented by Equation (2):(2)$$s=\sigma \left(C1D_{k}\left(y\right)\right)$$
where σ is the sigmoid activation function, y denotes the 1 × 1 × C feature Z being convolved, C1D denotes the one-dimensional convolution, and the size of the one-dimensional convolution kernel is indicated by *k*. *k* was obtained by Equation (3):(3)$$k=\phi \left(C\right)={\left|\frac{log_{2}\left(C\right)}{2}+\frac{1}{2}\right|}_{odd}$$
where C denotes the given channel dimension, and odd is the nearest odd number after taking the absolute value calculation.

3.Feature recalibration

The weight information of each channel obtained in Step 2 was multiplied by the corresponding original channel features, thereby achieving the goal of recalibrating the original feature information by enhancing the task-critical channel information in all the channels and suppressing the unimportant channel information. The operation could be represented by Equation (4):(4)$${\tilde{X}}_{c}=L_{c}\cdot X_{c}$$
where $L_{c}$ represents the weight coefficient of each channel, and $X_{c}$ represents the original channel feature information for each channel.

In this study, an improved lightweight channel attention mechanism was added to the CSP module of a CSPDarknet53-tiny network. This greatly improved the network’s ability to extract important feature information while reducing the number of parameters and computations to improve the accuracy of the network’s detection.

#### 3.3.2. The Improvement of the Feature Pyramid and Detection Network

In the traditional structure of a convolutional neural network, a fully connected layer is connected after the convolutional layer. Since the number of features in the fully connected layer is fixed, the size of the input image on the input side of the network is also fixed. In practical applications, the input image size is typically inadequate and must be cropped and stretched, which frequently distorts the image. Spatial pyramid pooling (SPP) [36] can generate fixed-scale features by processing input images of arbitrary sizes or scales and is robust to changes in the size and shape of an input image. Its structure is shown in Figure 4.

Inspired by the idea of SPP and YOLOv3-spp [37], this study improved a traditional SPP module, and the improved structure is shown in Figure 5. The structure consisted of five branches. The first branch connected the input directly to the output, the second branch downsampled the input through a maximum pooling of size 3 × 3 and then output, the third branch downsampled the input through a maximum pooling of size 5 × 5 and then output, the fourth branch downsampled the input through a maximum pooling of size 7 × 7 and then output, and the fifth branch downsampled the input through a maximum pooling of size 9 × 9 and then output. Since the step size of the pooling layer was 1 and the padding operation was performed before the pooling operation, the length, width, and depth of the feature map output from these five branches were the same. Finally, these five feature maps were concatenated. This dense SPP network was added after the backbone network since YOLOv4-tiny disregards the fusion of multiscale local region features on the same convolutional layer. This dense SPP network converted the 13 × 13 × 512 feature maps generated by the 15th convolutional layer into 13 × 13 × 2560 feature maps. This structure achieved the fusion between feature maps of local and global features, and the multiscale fusion enhanced the characterization ability of the feature maps so that more features were passed to the next layer of the network. The number of input feature maps was then reduced from 2560 to 256 using 1 × 1 convolution to extract useful features from the large number of relevant features, which were later pooled to different scales to improve the detection accuracy of traffic signs.

We enhanced the Yolo head module to increase the YOLOv4-tiny network’s ability to identify traffic signs. Following the enhanced feature extraction network, a YOLO detection layer was added. To create this detection layer, we fine-tuned a second YOLO detection layer and added convolutional layers with channel sizes of 128, 256, 512, and 24. The final output of this detection layer was a high-dimensional feature map of 52 × 52 × 24, which enhanced the accuracy of target localization and prediction. YOLOv4-tiny can only generate feature maps with the dimensions of 13 × 13 × 24 and 26 × 26 × 24. With these improvements, the TSR-YOLO algorithm achieved first YOLO layer outputting feature maps of 13 × 13 × 24, second YOLO detection layer outputting feature maps of 26 × 26 × 24, and third YOLO detection layer outputting feature maps of 52 × 52 × 24. This method could better detect long-distance traffic signs in complex scenarios and solve the problem of inaccurate localization and prediction of YOLOv4-tiny when locating small targets at a far distance. The network’s three YOLO detection layers were used to process and forecast the bounding boxes, objectness score, class predictions, and anchor boxes, where anchor boxes were used to identify the bounding boxes for each object in each class in the traffic sign recognition dataset. Because three detection layers were used and there were three classes of traffic signs in the dataset, the number of channels for each detection layer was calculated by the formula (class + 4 + 1) × 3 before designing each YOLO detection layer, and the channel size was set to 24. After completing the above improvements, the TSR-YOLO algorithm structure and the algorithm’s detailed network configuration are given in Figure 6.

### 3.4. Anchor Boxes Using K-Means++ Clustering

The original YOLOv4-tiny model’s anchor boxes were obtained by clustering the COCO dataset [38] and the Pascal VOC dataset [39]. By analyzing these datasets, we found that the targets in these datasets were more different in size and shape from those in the traffic sign dataset, and the background of the traffic sign dataset was more complex. As a result, the original anchor box size was unsuitable for Chinese traffic sign detection task and may harm the model’s training results. Figure 7 shows samples from the Pascal VOC dataset and the CCTSDB2021 dataset.

A typical k-means clustering algorithm [40] was used in the original YOLOv4-tiny model for a dimensional clustering analysis of training images to obtain prior boxes. Nevertheless, the randomness of the k-means algorithm for the selection of initial clustering centers may have detrimental impacts on the clustering effect. The k-means++ clustering algorithm was used instead of the k-means clustering algorithm in this work to improve the accuracy with which the proposed target detection network predicted a target’s location. The k-means++ clustering algorithm featured less clustering randomness, which reduced the bias of clustering results produced by the random selection of initial clustering centers. k-means++ was utilized to cluster the CCTSDB2021 dataset for Chinese traffic signs in order to generate more accurate and representative anchor boxes. The k-means++ technique for clustering works was as follows:


Determine the number of cluster centers *k* and the height and width set M of Chinese traffic signs in the given data.Choose one point randomly from the set M to satisfy the initial clustering center $q_{1}$.Determine D(x) the distance between each remaining point x in the set M and its nearest clustering center $q_{x}$. The greater the distance between the prior box and the next clustering center, the greater the probability P(x). This step should be repeated until *k* clustering centers are found.
(5)$$D\left(x\right)=1-IOU\left(x,q_{x}\right)$$
(6)$$P\left(x\right)=\frac{D{\left(x\right)}^{2}}{\sum_{x\in N}D{\left(x\right)}^{2}}$$$IOU\left(x,q_{x}\right)$ denotes the intersection ratio between the clustering center and each labeled box.Determine the distance *D(x)* between all points in the set M and the *k* cluster centers, and place the point in the cluster center category with the smallest distance. For the clustering results, recalculate each clustering category center $C_{i}$.
(7)$$C_{i}=\frac{\sum_{x\in C_{i}}x}{\left|C_{i}\right|}$$When the cluster center $C_{i}$ of each clustering category no longer changes, repeat Step 2 and output *k* cluster center results.


The CCTSDB2021 dataset was initially analyzed, and Figure 8 shows the annotated information of the samples in the dataset. The data for the clustering algorithm are in the annotated red box in Figure 8. Therefore, we give more detail about the data in the an-notated red box. The bndbox tag specifies the location of a traffic sign within an image, the xmin value specifies the horizontal coordinate of the upper-left corner of a traffic sign bounding box, and the ymin tag specifies the vertical coordinate of the upper-left corner of a traffic sign bounding box. The xmax tag specifies the horizontal coordinate of a sign’s lower-right corner. The ymax tag specifies the vertical coordinate of a bounding box’s lower right corner. 

The width and height of each traffic sign’s bounding box were used as horizontal and vertical coordinates. The width and height were normalized with respect to the original image to obtain the distribution of the actual boxes of traffic signs in the CCTSDB2021 dataset. Following the above k-means++ algorithm steps, a cluster analysis was then performed using the CCTSDB2021 dataset, with *k* set to 9, and nine clustering results were obtained for the CCTSDB2021 dataset, where the black plus sign is the cluster center, as shown in Figure 8. The nine clustering centers in Figure 9 were (0.009375, 0.01805556), (0.0140625, 0.02638889), (0.02109375, 0.0375), (0.018, 0.04857143), (0.022, 0.06285714), (0.03203125, 0.05448718), (0.029, 0.07714286), (0.042, 0.1), and (0.07533351, 0.14571429), respectively. The final parameters of anchor boxes needed to be transformed according to the original image size. When the input image size was 416 × 416, the coordinate values of the clustering centers were multiplied by 416 to obtain the nine clustering centers in the original image of (4, 8), (6, 11), (9, 16), (8, 20), (9, 26), (14, 23), (12, 32), (18, 42), and (32, 61). Large anchor boxes were used to predict big traffic signs, and small anchor boxes were used to predict small traffic signs. Thus, the prior boxes of (12, 32), (18, 42), and (32, 61) were used to predict the bounding box at the scale of 13 × 13; the prior boxes of (8, 20), (9, 26), and (14, 23) were used to predict the bounding box at the scale of 26 × 26; the remaining three prior boxes of (4, 8), (6, 11), and (9, 16) were used to predict the bounding box at the scale of 52 × 52.

### 3.5. Traffic Detection Using TSR-YOLO

The traffic sign detection process based on TSR-YOLO included dataset preprocessing, model training, and detection of traffic signs, as shown in Figure 10. First, the CCTSDB2021 training dataset was preprocessed to improve traffic sign detection performance and prevent model overfitting. In the training phase, the dataset was first loaded, and the anchor boxes were generated using the k-means++ clustering algorithm; the training parameters were then set, and the TSR-YOLO network model was initialized; finally, the weights of the TSR-YOLO model were iteratively updated using the loss function to converge the loss function and obtain the model weights for traffic sign detection. An image or video was input during the traffic sign detection phase, the trained model weights were loaded, and traffic signs were predicted. At this period, the obtained prediction results contained multiple prediction boxes that overlapped. Redundant prediction boxes were removed using the non-maximum suppression (NMS) algorithm, and the final test results were output.

## 4. Experimental Section

### 4.1. Dataset

In the current research on traffic sign detection and recognition, the algorithm performance was primarily evaluated using well-known public transportation sign datasets, such as the GTSDB (German Traffic Sign Detection Benchmark), the BTSD (Belgian Traffic Sign Dataset), and the STSD (Swedish Traffic Sign Dataset) [41]. The aforementioned datasets are limited to European traffic signs, the samples are gathered primarily under optimal lighting settings, and there are substantial discrepancies between Chinese and European traffic signs [42]. We used the CCTSDB2021 dataset, which was composed of 423 films from traffic recorders at varied times, locations, and weather conditions, to accomplish real-time detection of Chinese traffic signs in difficult circumstances. The dataset included street traffic scenes, high-speed traffic scenes, rain traffic scenes, evening traffic scenes, and backlight traffic scenes. The diversity and coverage of the dataset were well-ensured, which was more compatible with the task of Chinese traffic sign detection under complex scenarios.

According to Table 1, the traffic signs in the CCTSDB2021 dataset were divided into three categories based on their respective meanings: prohibitive signs, warning signs, and mandatory signs. The prohibitive signs had a white background, a red circle, a red bar, and a black pattern, and their shapes were a circle, octagon, or equilateral triangle with the top angle pointing downward. The warning signs had a yellow background, a black border, and a black pattern, and their shape was an equilateral triangle with the top angle pointing upward. The mandatory signs had a blue background and a white pattern, and their bodies were composed of a circle, a rectangle, or a square. The training set for this dataset had 16,356 images, with 13,876 prohibitive signs, 4598 warning signs, and 8363 mandatory signs. This dataset’s test set was composed of 1500 images, and the entire test set had 3228 traffic signs. In a ratio of 9:1, the training set was divided into a training set and a validation set.

### 4.2. Experiment Configuration

In this study, we built neural networks with the Pytorch deep-learning framework and trained them on a GPU server using the parameters in Table 2 for the experimental environment.

When training the network model in the above experimental setting, no pretrained model was used. The input image size of the network was set to 416 × 416, and the weight parameters of the convolutional neural network were optimized using a stochastic gradient descent (SGD) optimizer. The learning rate was adjusted using cosine annealing LR. The TSR-YOLO model was updated and optimized to obtain the optimal model after several stable iterations of learning on the training set images. The main parameter settings for model training are shown in Table 3.

### 4.3. Evaluation Metrics

Multiple evaluation criteria were employed to analyze the proposed modified Yolov4-tiny model from diverse angles in order to evaluate the detection performance of the algorithm in complex road scenarios more objectively. In this work, precision (P), recall (R), mean average precision (mAP), F-1 score, and frames per second (FPS) were utilized to evaluate the performance of the algorithm. FPS denoted the number of images processed per second in the evaluation metrics. While true positives (TPs) indicated that the meaning of the identified traffic signs matched their actual meaning, false positives (FPs) occurred when traffic signs were identified, but the results of the detection contradicted the actual meaning of the signs. False negatives (FNs) were missed traffic signs by the model, P(R) was a function with R as a parameter, and “classes” was the number of classes in the dataset. Average precision (AP) was the average accuracy of a single category’s detection result. It showed how well the model worked at detecting the category’s target. The *mAP* metric is often used to measure the accuracy of multicategory target detection. It was the average of the *AP* values of all the categories in the dataset and is one of the most important metrics for measuring how well target detection works. As a result, the metrics could be calculated using the equations below.
(8)$$Precision=\frac{TP}{TP+FP}$$
(9)$$Recall=\frac{TP}{TP+FN}$$
(10)$$AP=\int_{0}^{1}P\left(R\right)d\left(R\right)$$
(11)$$mAP=\frac{1}{classes}\sum_{i=1}^{classes}\int_{0}^{1}P\left(R\right)d\left(R\right)$$
(12)$$F-1Score=2*\frac{\left(Precision*Recall\right)}{\left(Precision+Recall\right)}$$

### 4.4. Experimental Results and Analyses

#### 4.4.1. Evaluation Results

Using the CCTSDB2021 public dataset, the TSR-YOLO method was evaluated and compared to the original YOLOv4-tiny algorithm to produce more intuitive results. Table 4 displays the experimental detection results, as well as the evaluation metrics of *AP*, precision, recall, F-1 score, and mAP for each category in the dataset.

The mAP value of the TSR-YOLO algorithm for the CCTSDB2021 dataset was 8.23% higher than that of the original YOLOV4-tiny algorithm, as shown in Table 4, indicating that the algorithm proposed in this study had a high detection accuracy. Meanwhile, the F-1 score, recall, and precision of the TSR-YOLO algorithm were 3.04%, 1.60%, and 5.02% higher than those of YOLOV4-tiny, respectively. In addition, for the *AP* values of each class of the dataset, the *AP* values of the proposed algorithm for the prohibitive traffic sign class, the warning traffic sign class, and the mandatory traffic sign class were 92.51%, 93.54%, and 92.11%, respectively, which were improved by 11.46%, 8.98%, and 4.25%, respectively, compared with the original algorithm. The detection accuracy of the algorithm for each traffic sign class was improved to varying degrees, especially the detection accuracy of prohibitive traffic signs, which was greatly enhanced. To further compare the *AP* values in more detail, the PR curves for each category of these two algorithms are shown separately in Figure 11. The *AP* value for each category of traffic signs was the region contained by the PR curve and the coordinate axes. The *AP* value was higher and the performance was improved when the area that the angle and coordinate axes covered was larger. The figure also demonstrates that the TSR-YOLO model achieved a high detection performance in all three traffic sign categories, with considerable improvements in each category compared to YOLOv4-tiny, indicating that it was more capable of accurately identifying traffic signs in difficult settings.

In order to evaluate the efficacy of the proposed algorithm, a relevant test was conducted in a natural scene using an in-car camera by selecting a video containing traffic signs for frame extraction and processing, processing the video into multiple images, and evaluating the processed images with the algorithm. Red identification boxes indicate prohibitive traffic signs, blue identification boxes represent mandatory traffic signs, and green identification boxes represent warning traffic signs in Figure 12 and Figure 13.

These test images were captured on urban roads, as shown above, and were consistent with detecting traffic signs in complex scenarios. The detection precision of the algorithm described in this study was extremely high, and it achieved wide adoption.

These test images were captured on the highway, as shown above, and the improved YOLOv4-tiny algorithm achieved full recognition with a very high detection accuracy.

In conclusion, the method presented in this work could detect traffic signs in complicated environments.

#### 4.4.2. Performance Comparison

The algorithm described in this study was thoroughly validated by comparing it to advanced traffic-sign-detecting methods using the CCTSDB2021 dataset. Multiple evaluation criteria were used in this experiment to perform a quantitative, all-around evaluation from many different points of view. The results of the comparison are shown in Table 5. 

The results of the detection are shown in Table 5. First, Faster R-CNN is a two-stage detection model that is relatively new. The algorithm had a high detection accuracy and could detect traffic signs with precision, but its model was relatively vast and its detection speed was relatively slow. The YOLOv3, SSD, and YOLOv4 algorithms were the most representative one-stage model algorithms. Chen et al. proposed a more advanced T-YOLO based on Yolov3, and from the table, we can see that the *mAP* value of this algorithm reached up to 97.30%, but the FPS value of this algorithm was only 19.30. Shan et al. changed an SSD model to further improve the algorithm’s detection. Ren et al. combined a classical MobileNetv2 network with an SSD algorithm, which significantly improved the detection accuracy and speed, and the *mAP* value of YOLOv4 reached 95.8%; the size of this model was 243.94 MB. It can be concluded that the detection accuracies of these models were relatively high. Nevertheless, the models were typically very large, and the detection speeds were slow, making them unsuitable for edge devices on smart automobiles in complicated scenarios for real-time detection. Yolov4-tiny is a great, light-weight detection model, and the speed of detection and the size of the model were better-suited for real-time detection on edge devices. However, this technique had low detection accuracy. TSR-YOLO combined several optimization modules and improved the YOLOv4-tiny algorithm in terms of detecting traffic signs. In conclusion, the algorithm described in this research surpassed prior algorithms by balancing detection accuracy, detection speed, and model size and could better match the requirements of intelligent vehicle-sensing systems for the real-time detection of complicated road environments.

The TSR-YOLO algorithm was tested with YOLOv4-tiny in four complex environments selected from the CCTSDB2021 dataset, including a well-lit environment, a night environment, a rainy environment, and a snowy environment. The test results are depicted in Figure 14, Figure 15, Figure 16 and Figure 17, where “prohibitory” represents prohibitive traffic signs, “warning” represents warning traffic signs, and “mandatory” represents mandatory traffic signs. (a) and (c) represent the detection results of TSR-YOLO, whereas (b) and (d) represent the detection results of YOLOv4-tiny. Figure 14 shows that the method described in this paper worked well in places where there was enough light. In the first figure, the TSR-YOLO algorithm’s detection accuracy for prohibitive traffic signs was 100% and 58%, while YOLOv4-tiny’s detection accuracy was 98% and one of the traffic signs was not detected. The algorithm shown in this study was 14% more accurate than YOLOv4-tiny’s detection accuracy, and the improvement was observable. The second figure shows that the TSR-YOLO algorithm and YOLOv4-tiny algorithm detected 100% of the warning traffic signs, but the YOLOv4-tiny algorithm incorrectly identified the background as a traffic sign. The algorithm in this work had more advantages in an environment with ideal lighting conditions. In a night environment, the detection results are shown in Figure 15.

As shown above, the TSR-YOLO algorithm had a higher detection accuracy in the nighttime scenario than the YOLOv4-tiny method. However, the detection results of the two methods did not differ much, with an error rate of less than 3%. Figure 16 displays the results for detection on rainy days.

Compared to the YOLOv4-tiny method, the detection results of the TSR-YOLO algorithm in rainy conditions were increased by around 7%, which considerably enhanced the detection accuracy. From the first image, the algorithm in this paper had more accurate target localization than YOLOv4-tiny in rainy environments. Detection results in a snowy climate are shown in Figure 17.

The detection results of the TSR-YOLO algorithm differed greatly from those of the YOLOv4-tiny method, as depicted in the images above. The YOLOv4-tiny algorithm had a scenario of leakage detection with a relatively low detection rate. The technique presented in this research improved the detection accuracy by approximately 20% compared to the YOLOv4-tiny algorithm, and there was no leakage detection, which improved the accuracy of traffic sign detection. In a snowy environment, the TSR-YOLO algorithm outperformed the YOLOv4-tiny method.

The above results demonstrate that TSR-YOLO had a significant effect on detection accuracy in normal lighting conditions, increasing it by 14%. In a snowy environment, the effect of the 20% boost was astounding. Even though TSR-YOLO did not improve the detection effect as dramatically as on sunny days and in snowy environments, it still improves the detection effect by 7% on rainy days. Due to noises such as rain, the detection results showed that TSR-YOLO could find traffic signs more accurately than the original algorithm. This shows that the algorithm was more robust. In a dark environment, the majority of the image backgrounds were black, background interference was minimal, and due to the effect of light, the traffic signs were clearer and easier to detect. Thus, the detection accuracies of TSR-YOLO and YOLOv4-tiny were nearly equivalent, and the rate of accuracy for each was close to 100%. In conclusion, the improved network was more adaptable to a complex natural environment and had improved localization and recognition accuracy.

## 5. Discussion

Compared to the YOLOv4-tiny algorithm, the TSR-YOLO algorithm significantly improved the detection accuracy for Chinese traffic sign recognition in complex scenarios, but the improved method still had limitations. First, TSR-YOLO could only roughly recognize traffic signs in three categories—warning, prohibitive, and mandatory—without fine-grained division, which was insufficient for scenarios requiring more precise traffic sign detection results. Second, this study only performed tests in four situations: a well-lit environment, a night environment, a rainy environment, and a snowy environment. It did not take into account all natural situations, such as other extreme weather conditions and conditions where traffic signs are blocked, faded, or broken. For future research, we plan to optimize the TSR-YOLO model to make it suitable for environments with greater complexity.

## 6. Conclusions

An enhanced traffic sign detection algorithm based on YOLOv4-tiny was proposed to address the issue of the low accuracy of current lightweight networks at detecting traffic signs in complex circumstances. Some improvement strategies were offered based on YOLOv4-tiny. The k-means++ clustering technique first built suitable anchor boxes for a traffic sign dataset. A BECA module was then implemented to improve the model’s ability to extract essential feature information in response to the fact that the extracted characteristics of the backbone network were mostly concentrated on a CSP module. In addition, a dense SPP module was added to the upgraded feature extraction network so that the convolutional neural network could fuse local and global features more effectively. Lastly, a Yolo detecting layer was added to more precisely detect and localize small targets at a great distance in a complicated environment, hence enhancing the algorithm’s detection performance and achieving improved detection results. The experiments showed that, for the CCTSDB2021 dataset, the algorithm described in this paper was faster and more accurate than both YOLOv4-tiny and other excellent models. Therefore, the suggested network was more suited for real-time traffic sign detection on edge terminals deployed in intelligent vehicle-driving systems.

## Figures and Tables

**Figure 1 sensors-23-00749-f001:**
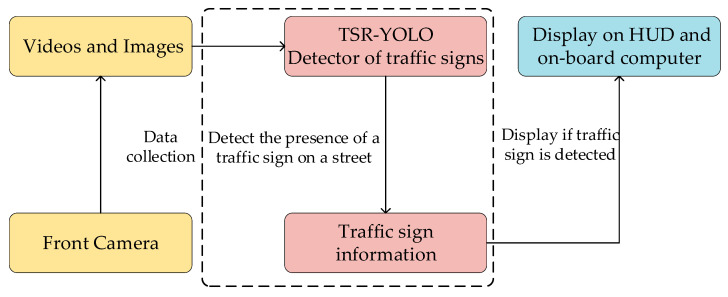
Traffic sign recognition system.

**Figure 2 sensors-23-00749-f002:**
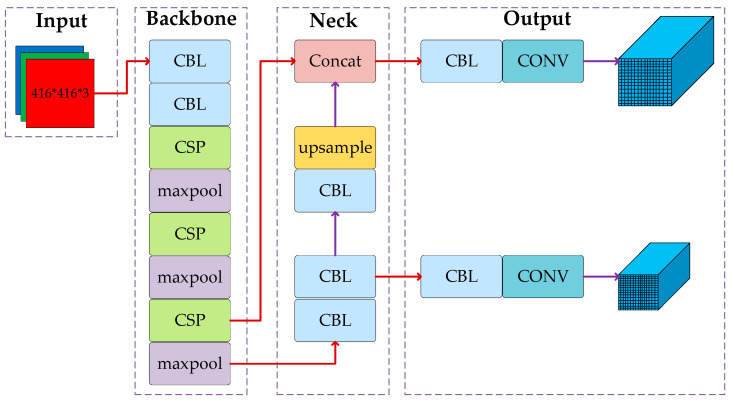
Structure of the YOLOv4-tiny network.

**Figure 3 sensors-23-00749-f003:**
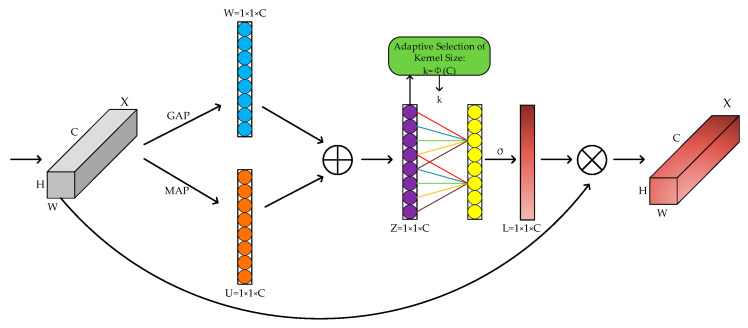
BECA structure.

**Figure 4 sensors-23-00749-f004:**
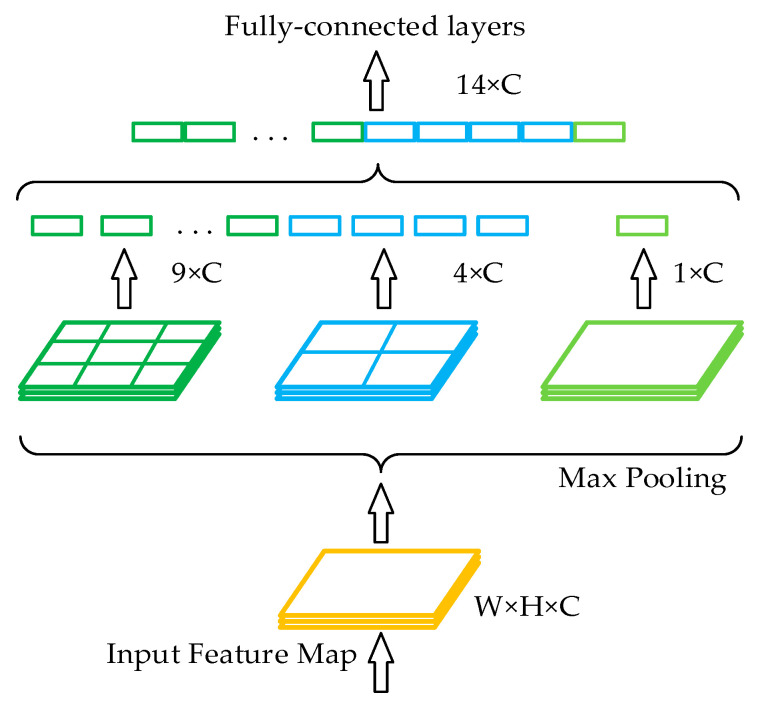
Spatial pyramid pooling.

**Figure 5 sensors-23-00749-f005:**
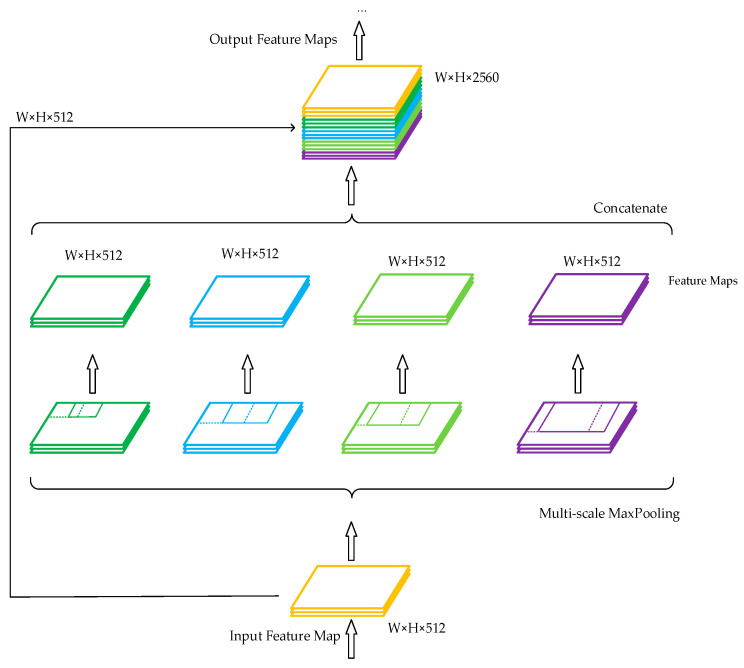
Improved dense spatial pyramid pooling.

**Figure 6 sensors-23-00749-f006:**
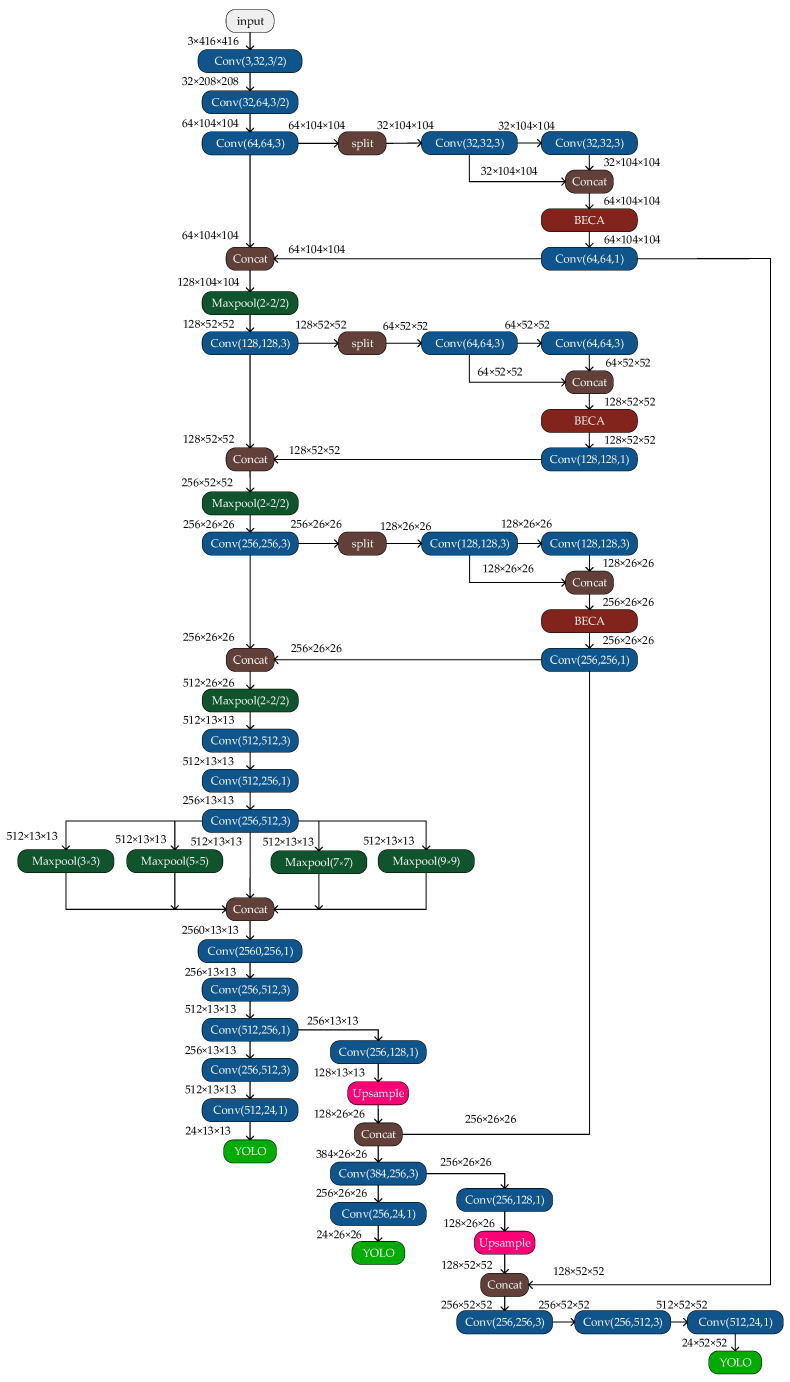
Improved network TSR-YOLO.

**Figure 7 sensors-23-00749-f007:**
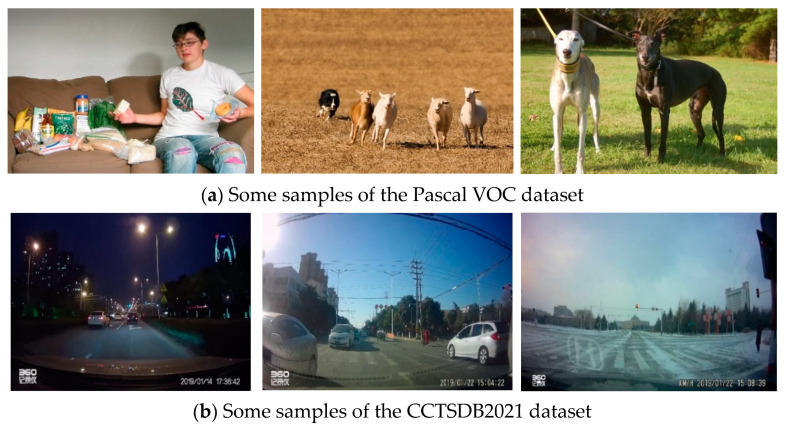
Some samples in the Pascal VOC and CCTSDB2021 datasets.

**Figure 8 sensors-23-00749-f008:**
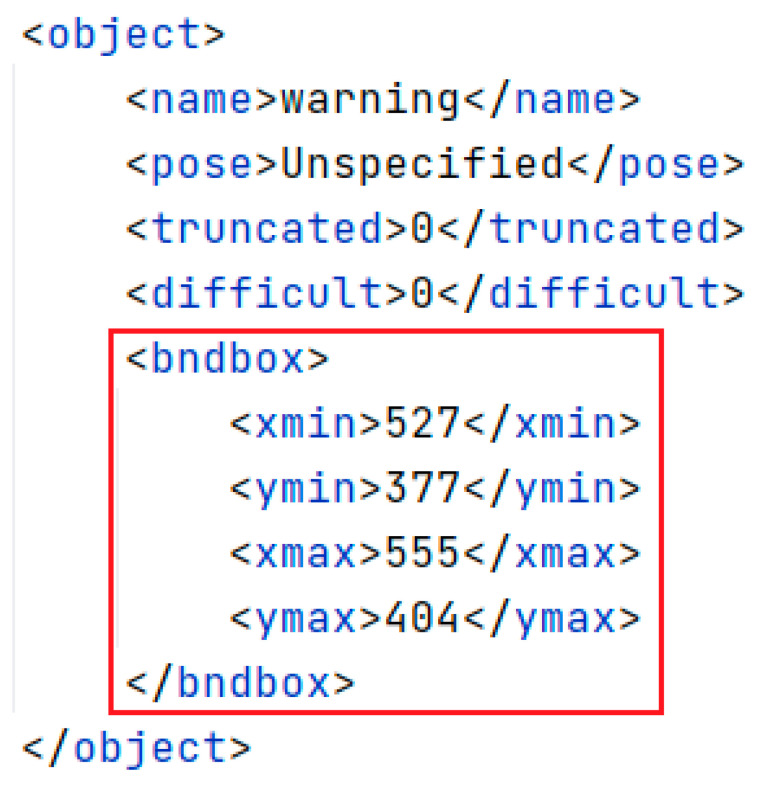
Annotation information of the image.

**Figure 9 sensors-23-00749-f009:**
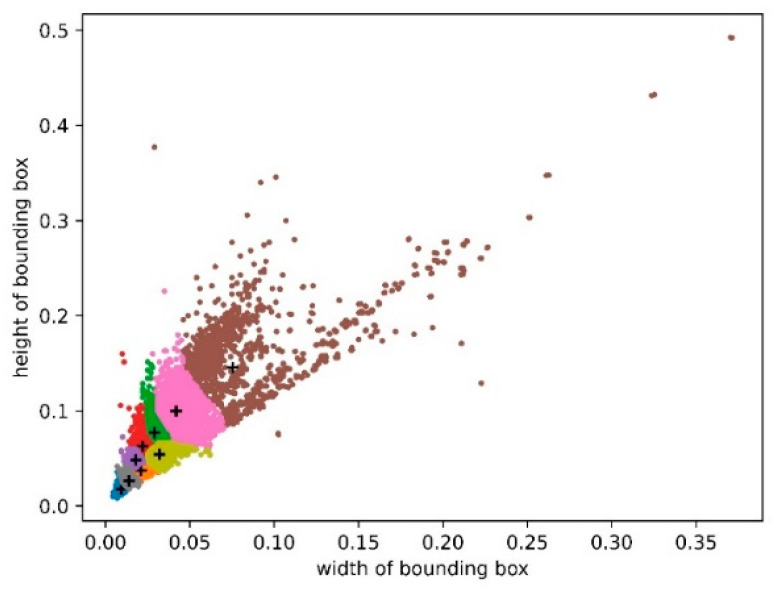
Distribution of clustering centers in the CCTSDB2021 dataset.

**Figure 10 sensors-23-00749-f010:**
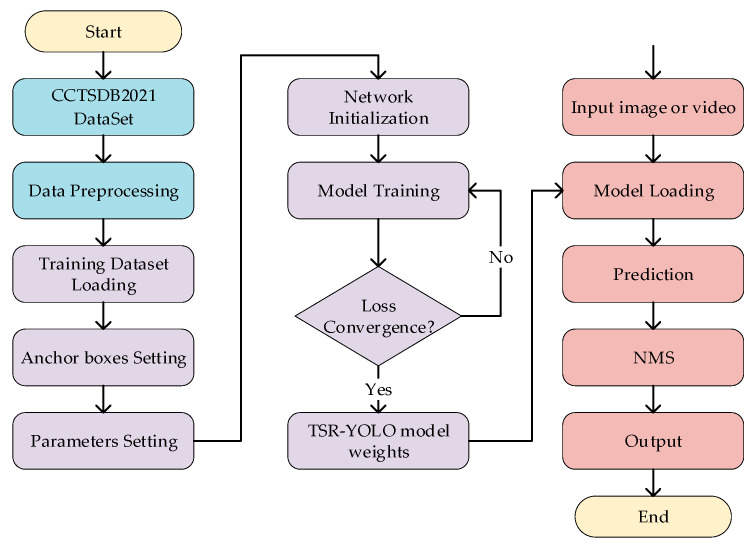
The traffic detection process based on TSR-YOLO.

**Figure 11 sensors-23-00749-f011:**
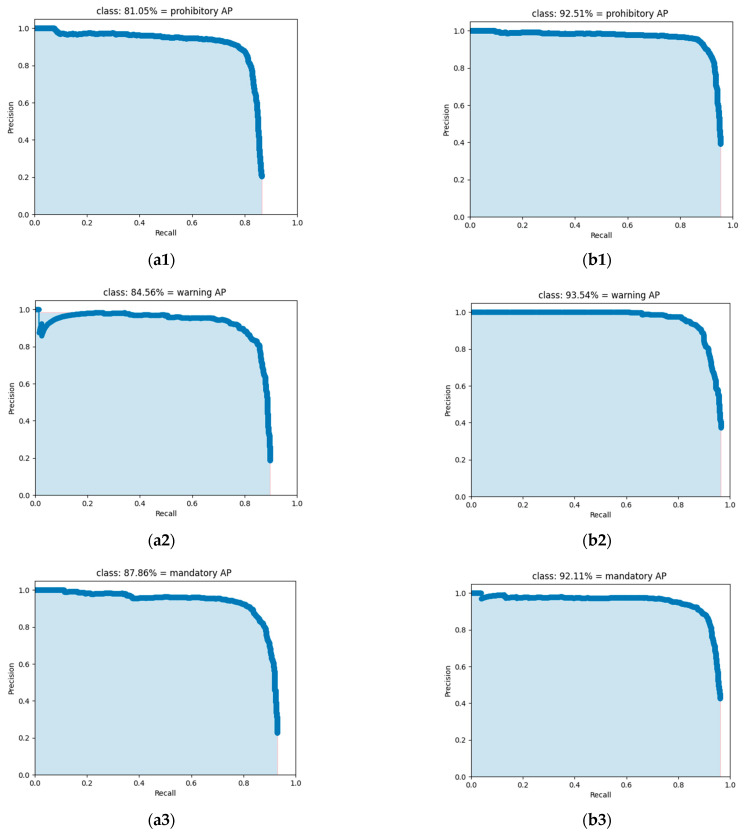
Comparison of PR curves for different categories, where (**a1**–**a3**) are the PR curves for each type obtained by the YOLOv4-tiny algorithm, and (**b1**–**b3**) are the PR curves for each type accepted by the TSR-YOLO model.

**Figure 12 sensors-23-00749-f012:**
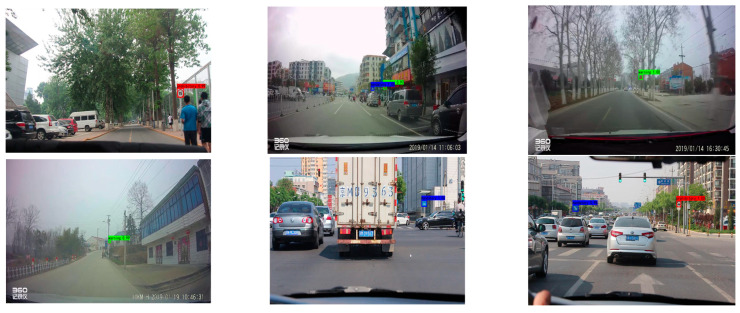
Urban road test results.

**Figure 13 sensors-23-00749-f013:**
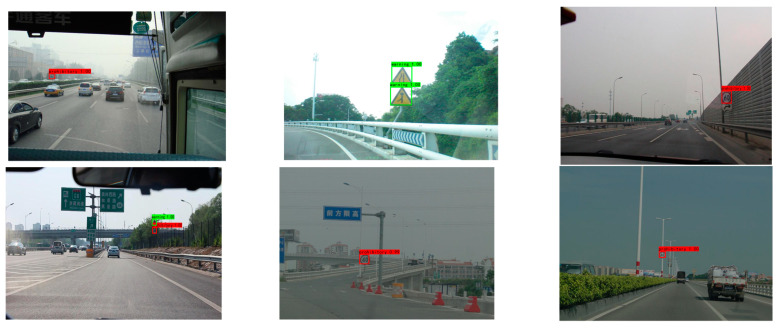
Highway test results.

**Figure 14 sensors-23-00749-f014:**
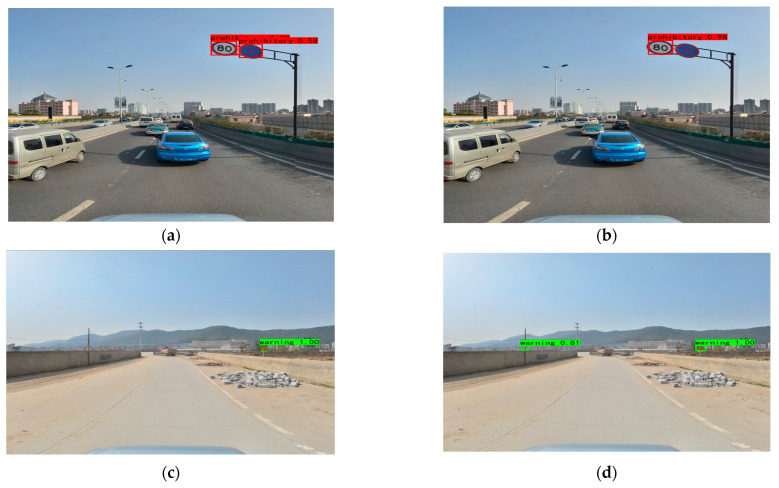
Comparison results of different algorithms in excellent lighting conditions, where (**a**,**c**) are the results of TSR-YOLO, (**b**,**d**) are the results of YOLOv4-tiny.

**Figure 15 sensors-23-00749-f015:**
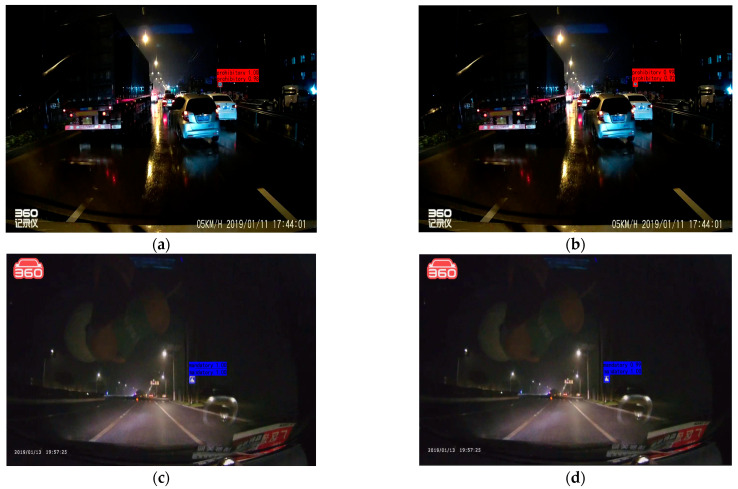
Comparison results of different algorithms in a night scenario, where (**a**,**c**) are the results of TSR-YOLO, (**b**,**d**) are the results of YOLOv4-tiny.

**Figure 16 sensors-23-00749-f016:**
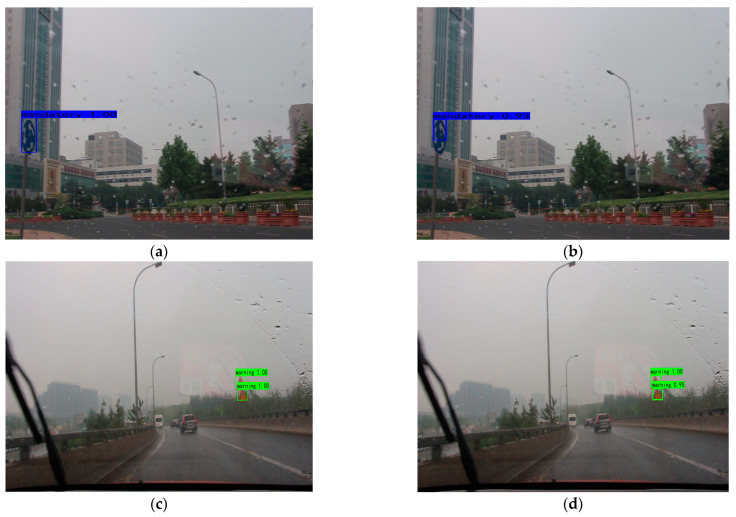
Comparison results of different algorithms in rainy weather, where (**a**,**c**) are the results of TSR-YOLO, (**b**,**d**) are the results of YOLOv4-tiny.

**Figure 17 sensors-23-00749-f017:**
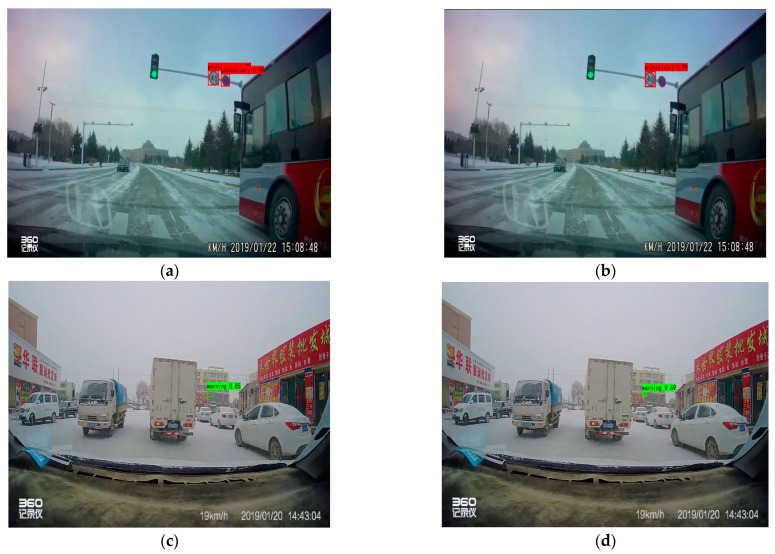
Comparison results of different algorithms in snowy weather, where (**a**,**c**) are the results of TSR-YOLO, (**b**,**d**) are the results of YOLOv4-tiny.

**Table 1 sensors-23-00749-t001:** Selected examples of three types of traffic signs in the CCTSDB2021 dataset.

Prohibitory	Warning	Mandatory
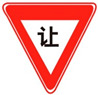	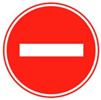	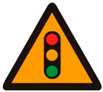	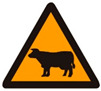	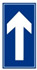	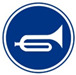
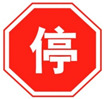	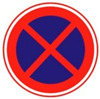	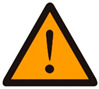	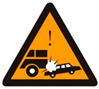	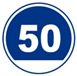	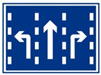
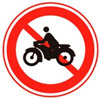	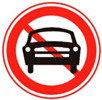	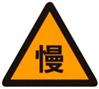	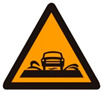	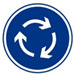	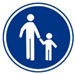

**Table 2 sensors-23-00749-t002:** Experimental environmental parameters.

Experimental Environment	Environment Configuration
Operating system	Windows11
CPU	Intel(R) Core (TM) i7-10750 H CPU @ 2.60 GHz
GPU	NVIDIA GeForce RTX 2060
Programming language	Python 3.10
Deep-learning framework	Pytorch 1.12
Acceleration platform	CUDA11.3;cuDNN8.2

**Table 3 sensors-23-00749-t003:** Experimental parameters of network training.

Attribute	Value
epoch	500
batch size	16
initial learning rate	0.001
momentum	0.937
weight_decay	0.0005
input shape	(416, 416)
mosaic	true
mixup	true
lr_decay_type	cos

**Table 4 sensors-23-00749-t004:** Performance of the improved YOLOv4-tiny ablation results for the CCTSDB2021 dataset.

Network	Class		Evaluation Indicator			
		*AP* (%)	Precision (%)	Recall (%)	F-1 Score (%)	*mAP* (%)
YOLOv4-tiny	prohibitory	81.05	91.60	78.13	84.33	84.49
	warning	84.56				
	mandatory	87.86				
Proposed	prohibitory	92.51	96.62	79.73	87.37	92.72
	warning	93.54				
	mandatory	92.11				

**Table 5 sensors-23-00749-t005:** Detection results of different networks for CCTSDB dataset.

Model	Evaluation Indicator				
	P/%	R/%	mAP@0.5/%	Speed (fps)	Size (MB)
Shan et al. [31]	-	-	85.00	-	-
Chen et al. [32]	91.30	-	97.30	19.30	-
Ren et al. [43]	-	-	93.20	45.00	-
YOLOv4 [44]	88.10	92.80	95.80	-	243.94
Faster R-CNN [44]	91.60	90.70	93.50	21.70	-
YOLOv4-tiny	91.60	78.13	84.49	112.69	23.40
Ours	96.62	79.73	92.72	80.55	41.37

## Data Availability

The data that support the findings of this study are openly available in CCTSDB 2021 at https://doi.org/10.22967/HCIS.2022.12.023.

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
