# Peer review of "TSR-YOLO: A Chinese Traffic Sign Recognition Algorithm for Intelligent Vehicles in Complex Scenes"

_sensors, 2023, doi:10.3390/s23020749_

Round 1

Reviewer 1 Report

This paper focuses on developing a traffic sign recognition system based on YOLO real-time object detection system. The paper is well written and I think it will have a significant impact in this field. The authors should only explain the evaluation metrics in (10)-(11) in detail.

Reviewer 2 Report

Further explanation is necessary about the coordinate of anchor boxes, where the numbers are presented without a description of relevant data(images) of CCTSDB2021. For an example, the sufficient explanation would be helpful for explaining Figure 8.

The experiment configuration (4-2) doesn’t include the reasonable performance demonstration of the authors’ claim in Abstract, where the TSR-YOLO achieved the successful balance between detection performance and deployment difficulty in embedded devices, enabling deployment in intelligent vehicles with limited computational resources. However, there isn’t a rigorous analysis about the real time operation or required computational resources of embedded devices.

It is unlikely enough to accept that the TSR-YOLO has the better performance than other algorithms, just with a comparison based on only Chinese traffic sign recognition. The experimental results and analysis (4.4) is vague or limited to support the algorithm’s favorable effect on technological advancement in the statement of Abstract.

Reviewer 3 Report

>> The language usage throughout this paper need to be improved, the author should do some proofreading on it. Give the article a mild language revision to get rid of few complex sentences that hinder readability and eradicate typo errors.

>> Your abstract does not highlight the specifics of your research or findings. Rewrite the Abstract section to be more meaningful. I suggest to be Problem, Aim, Methods, Results, and Conclusion.

>> Introduction section can be extended to add the issues in the context of the existing work and how proposed algorithms/approach can be used to overcome this.

>> The problems of this work are not clearly stated. There is ambiguity in statement understanding.

>> More clarifications and highlighted about the research gabs in the related works section.

>> identified research gaps and contribution of the proposed study should be elaborated.

>> update the related works by discussed the following studies:

>> I feel that more explanation would be need on how the proposed method is performed.

>> If no one has proposed before a method like the proposed algorithm, this claim should be highlighted much more. Else, it should be indicated who has done this, and it should be indicated what the innovations of the current paper are.

>> Authors should add the parameters of the algorithms.

>> A comparison with state of art in the form of table should be added

>> Results need more explanations. Additional analysis is required at each experiment to show the its main purpose.

>> The Limitations of the proposed study need to be discussed before conclusion.